# Elevated expression of TUBA1C in breast cancer predicts poor prognosis

Yi Zhao[1], Wenwen Wang[1,2], Jinming Li[1,3], Jiarui Du[1], Qiqi Xie[1], Miaozhou Wang[1], Zhen Liu[1], Xingfa Huo[1], Fuxing Zhao[1], Dengfeng Ren[1], Jiuda Zhao[1]*, GuoShuang Shen[1]*

1 Breast Disease Diagnosis and Treatment Center of Affiliated Hospital of Qinghai University & Affiliated Cancer Hospital of Qinghai University, Xining, QingHai, China, 2 Department of Pharmacogenomics, State Key Laboratory of Cancer Biology, Fourth Military Medical University, Xi'an, Shaanxi, China, 3 Graduate School of Qinghai University, Xining, QingHai, China

☯ These authors contributed equally to this work.
* jiudazhao0519@163.com (JZ); guoshuangshen6688@163.com (GSS)

**Data Availability Statement:** We have performed the upload of the minimum dataset according to the PLOS ONE journal's request for our manuscript, which we uploaded at the following website address: https://datadryad.org/stash/

## Abstract

α1C-tubulin (*TUBA1C*) is a member of the α-tubulin family and has served as a potential biomarker in a variety of cancers in many studies. In this study, the gene expression profile of *TUBA1C* in The Cancer Genome Atlas (TCGA) was extracted for analysis, and the prognostic value of *TUBA1C* in breast cancer was comprehensively evaluated. The Wilcoxon signed-rank test, Kruskal-Wallis test, and logistic regression analysis were performed to confirm the correlations between *TUBA1C* expression and the clinical characteristics of breast cancer patients. The effect of *TUBA1C* expression on the survival of breast cancer patients was assessed by Kaplan-Meier curve, Cox regression analysis, and the Kaplan-Meier plotter (an online database). The TCGA data set was used for the Gene Set Enrichment Analysis (GSEA). The results confirmed that high *TUBA1C* expression in breast cancer was closely correlated with survival time, survival status, and tumor size. In addition, elevated *TUBA1C* expression can predict poor overall survival (OS), recurrence-free survival (RFS), and distant metastasis-free survival (DMFS). Univariate and multivariate analyses (Cox regression analyses) confirmed that *TUBA1C* was an independent prognostic factor for the OS of breast cancer patients. The GSEA identified that the high *TUBA1C* expression phenotype was differentially enriched in cell cycle, basal transcription factor, P53 signaling pathway, pathways in cancer, TOLL-like receptor signaling pathway, and NOD-like receptor signaling pathway. In summary, high messenger RNA (mRNA) expression of *TUBA1C* is an independent risk factor for poor prognosis of breast cancer.

## Background

Breast cancer is a common malignancy in women worldwide, as new breast cancer cases account for 11.67% of all new cancer cases each year, and of these, breast cancer mortality accounts for 6.69% of all cancer deaths [1]. Therefore, breast cancer seriously threatens the lives and health of women. In recent years, with continuous advancements in medical technology, considerable progress has been made in the early diagnosis and treatment of breast

share/vxLw66WOaURhhfMojF44xMguUP9YtgY_eBGDtXZu3fw.

**Funding:** this work was supported by grants from the Key Research & Development and Transformation Project of Qinghai Province for 2018 (2018-SF-113).

**Competing interests:** The authors have declared that no competing interests exist.

cancer, which has led to some improvements in prognosis. Unfortunately, many patients still cannot be diagnosed early and are at risk for recurrence and metastasis due to a lack of more sensitive and specific prognostic indicators [2]. The existing pathological staging and molecular subtypes of breast cancer do not provide an accurate patient prognosis, and more prognostic markers are needed to reflect the diversity of tumor subtypes, improve patient risk stratification, and adjust individualized treatment strategies.

As an important component of the cytoskeleton, microtubules, which are composed of tubulin, have a plus-end and a minus-end, each of which has a different function [3,4]. At present, seven tubulin subtypes have been confirmed, each of which also has a distinct function [5]. Microtubules can participate in cell proliferation, intracellular transport of substances, and signal transduction by means of polymerization and depolymerization, and thus, they maintain normal cell morphology [6]. Microtubules also play an important role in cell division and chromosome segregation. Intracellular microtubules are primarily reticulate or in bundles and interact with other proteins in these two forms to participate in the formation of many important structures, including cell spindles, flagella, and cilia, whereas α-tubulin is one of the main subtypes that forms microtubule structures [7,8]. Recently, substantial evidence has indicated that α1C-tubulin (*TUBA1C*), which is a component of microtubules, is closely related to the occurrence and development of a variety of cancers [9–12]. For example, abnormally elevated *TUBA1C* expression in pancreatic cancer cells is associated with the prognosis of pancreatic cancer patients [10]. Although the exact mechanisms of *TUBA1C* in disease are still unclear, existing research suggests that *TUBA1C* may be a powerful potential prognostic marker of cancer progression and metastasis.

Observation of the gene expression profile suggests that *TUBA1C* might play an important role in breast cancer [13,14]. However, the correlations between abnormally elevated *TUBA1C* expression and breast cancer prognosis as well as other clinical factors of breast cancer have not been clearly elucidated. In this study, sequencing data, clinical information, and follow-up data for patients were extracted from The Cancer Genome Atlas (TCGA) to evaluate the differential expression of *TUBA1C* between breast cancer patients and healthy individuals, after which a pairwise comparison was performed. In addition, after patients were divided into *TUBA1C* high and low expression groups, the correlations between different *TUBA1C* expression levels and overall survival (OS), recurrence-free survival (RFS), distant metastasis-free survival (DMFS), post progression survival (PPS), and other clinical characteristics of breast cancer patients were analyzed, and gene set enrichment analysis (GSEA) was used to further explore the biological pathways regulated by *TUBA1C*. The results demonstrated that *TUBA1C* is a potential prognostic biomarker of breast cancer.

## Methods

### 2.1 Ethical statement

This study was approved by the Ethics Committee of Qinghai University Affiliated Hospital. All experimental data were derived from public databases, thus ensuring that informed consent was obtained for all data used in the study.

### 2.2 RNA sequencing (RNA-seq) gene data for patients and bioinformatics analysis

The gene expression data in this study and the corresponding clinical patient data were obtained from the TCGA database(TCGA, http//gdc.cancer.gov/) [15]. After exclusion of incomplete data, the RNA-seq gene expression data and the corresponding clinical data for

1085 breast cancer patients were collected. The differential expression, correlation analysis of clinical characteristics, univariate Cox analysis, multivariate Cox analysis, and logistic regression analysis were performed using R software (version 4.0.3).

## 2.3 Gene Expression Profiling Interactive Analysis (GEPIA) dataset

GEPIA (http://gepia.cancer-pku.cn/) is a new advanced interactive web server for analyzing RNA-seq gene expression data, including data from 9736 tumor samples and 8587 normal samples [16]. The included samples are all from the TCGA database and the Genotype Tissue Expression (GTEx) project. GEPIA has a variety of analytical functions, such as online analysis of differential expression between tumor and normal tissues, survival analysis, analysis based on different cancers or pathological stages, and the ability to search for similar genes. In addition, we used the limma packages, beeswarm packages in R language to further analyse the differential expression of the TUBA1C gene in breast cancer and normal breast tissue.

## 2.4 Kaplan-Meier plotter

The Kaplan-Meier plotter (http://kmplot.com/analysis/) is a prognosis-related online analysis tool, which was used to analyze the prognostic value of the *TUBA1C* gene in breast cancer tissues [17]. To analyze the prognostic indicators, i.e., OS, PPS, RFS, and DMFS, of breast cancer patients, breast cancer tissues were divided into high expression and low expression groups according to the median expression of *TUBA1C* messenger RNA (mRNA) and were evaluated using the Kaplan-Meier plotter. A p value < 0.05 indicated statistical significance.

## 2.5 GSEA

GSEA is an analysis tool for whole-genome expression microarray data that can construct a molecular signature database based on information about gene location, function, and biological significance [18]. Hybridization data of the expression profiles of a set of genes in two biological states were analyzed to determine statistical significance. In this study, raw data were processed in batches using GSEA to analyze the signaling pathways involved in the *TUBA1C* high expression group and the *TUBA1C* low expression group. *TUBA1C* expression was identified using phenotypic markers. The nominal p value and normalized enrichment score (NES) were used to sort the enriched pathways, with 1000 sorts per analysis.

## 2.6 Statistical analysis

The correlations between *TUBA1C* expression and OS, PPS, DMFS, and RFS were determined using the Kaplan-Meier plotter, and other statistical analyses were completed using R software (version 4.0.3). The Wilcoxon signed-rank test, Kruskal-Wallis test, and logistic regression analysis were performed to analyze the correlations between *TUBA1C* expression and the clinical characteristics of patients in the TCGA database. The median expression of *TUBA1C* mRNA was used to divide patients into the high and low expression groups. Univariate Cox analysis was used to analyze potential prognostic factors. Multivariate Cox analysis was performed to verify the correlations between *TUBA1C* expression and clinicopathological features as well as survival. P < 0.05 was considered statistically significant.

## Results

### 3.1 Characteristics of the study population

The analysis process of this study is shown in Fig 1. The clinical data for 1085 breast cancer patients were downloaded from the TCGA database and included patient age, survival status,

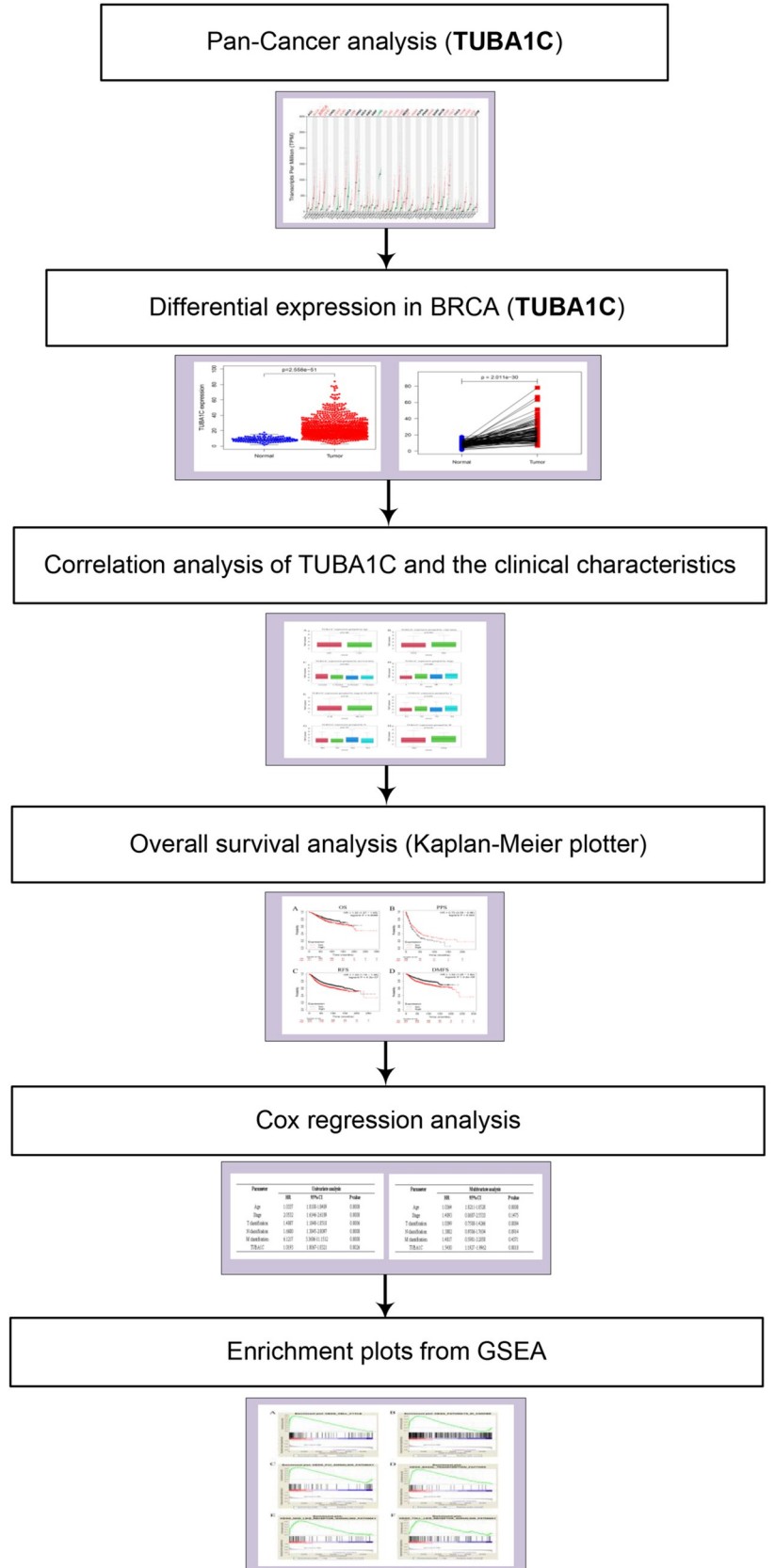

**Fig 1. Analysis workflow of this study.**

**Table 1. TCGA breast cancer patient characteristics.**

| Characteristic | Total(1085) | % |
|---|---|---|
| Age | | |
| <65 years | 746 | 68.76 |
| ≥65 years | 339 | 31.24 |
| Stage | | |
| I | 183 | 16.87 |
| II | 613 | 56.50 |
| III | 246 | 22.67 |
| IV | 19 | 1.75 |
| Not available | 24 | 2.21 |
| Gender | | |
| Female | 1085 | 100 |
| Male | 0 | 0 |
| Survival status | | |
| Survival | 937 | 86.36 |
| Death | 148 | 13.64 |
| T classification | | |
| T1 | 280 | 25.81 |
| T2 | 625 | 57.60 |
| T3 | 138 | 12.72 |
| T4 | 39 | 3.59 |
| TX | 3 | 0.28 |
| N classification | | |
| N0 | 512 | 47.19 |
| N1 | 358 | 33.00 |
| N2 | 119 | 10.97 |
| N3 | 76 | 7.00 |
| NX | 20 | 1.84 |
| M classification | | |
| M0 | 902 | 83.13 |
| M1 | 21 | 1.94 |
| MX | 162 | 14.93 |

clinical stage, tumor size, lymph node status, and the presence or absence of distant organ metastasis (Table 1).

## 3.2 *TUBA1C* expression is significantly increased in breast cancer

By downloading RNA-seq data from the TCGA database, differences in *TUBA1C* expression between breast cancer and normal breast tissues were statistically analyzed. The results showed that compared with normal breast tissue, *TUBA1C* expression was significantly increased in breast cancer tissues (P = 2.558e-51) (Fig 2B). In addition, pairwise difference analysis was performed on cancer and paracancerous tissues from the same sample taken from the TCGA database, and the results showed that *TUBA1C* expression in cancer tissues was significantly higher than that in paracancerous tissues (P = 1.446e-05) (Fig 2C). To further verify the analysis results, the changes in *TUBA1C* expression in different cancers were analyzed online through the GEPIA server, and it was found that *TUBA1C* expression was elevated in multiple cancer tissues including breast cancer (Fig 2A).

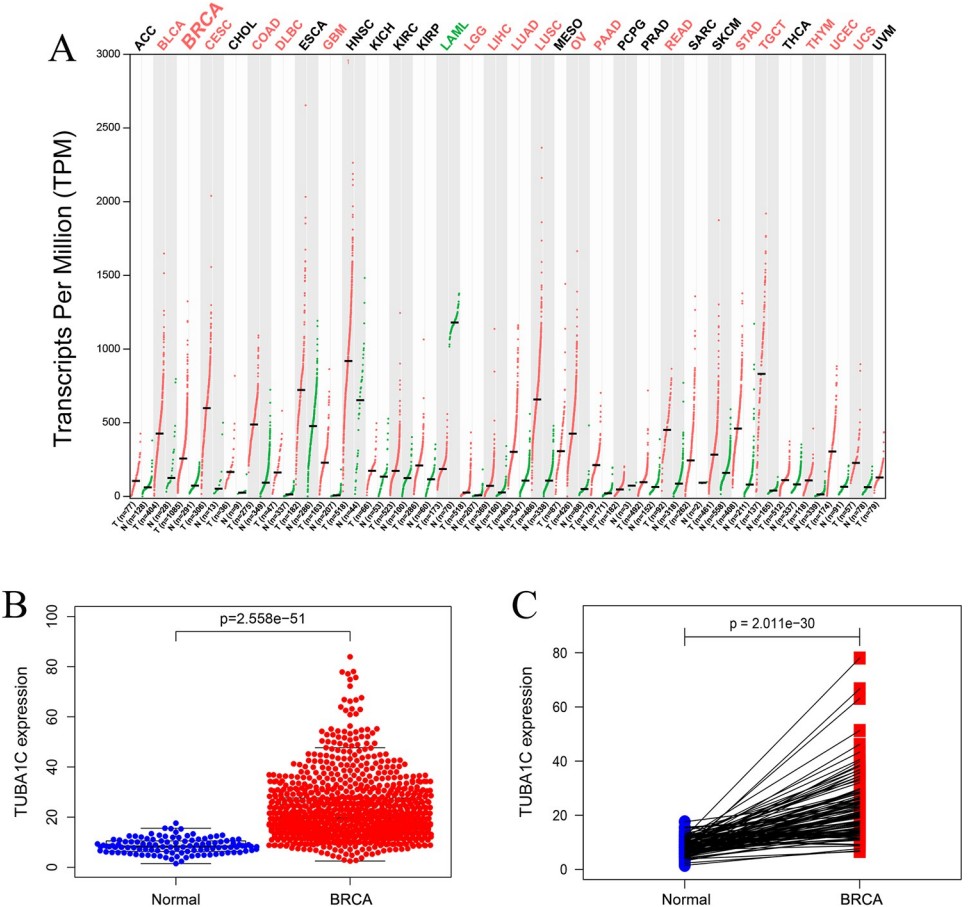

**Fig 2. mRNA expression of *TUBA1C* in different human tissues.** A. Expression pattern of *TUBA1C* in 33 types of tumor tissues and paired paracancerous tissues. B. Differences in *TUBA1C* expression between breast cancer tissues and normal breast tissues. C. Pairwise difference analysis of *TUBA1C* in breast cancer tissues and paired paracancerous tissues. Data were obtained from the GEPIA and TCGA databases.

### 3.3 Correlation analysis of *TUBA1C* and the clinical characteristics of breast cancer patients

The TCGA data included 1085 breast cancer samples with information on *TUBA1C* expression. The correlation analysis between *TUBA1C* expression and the clinical characteristics of these samples showed that high *TUBA1C* expression was correlated with survival time ($P = 0.032$), survival status ($P = 0.043$), and tumor size ($P = 0.005$) (Fig 3B, 3C and 3F). The logistic regression analysis found that *TUBA1C* was significantly correlated with the survival status and survival time of patients ($P < 0.05$) (Table 2). These results suggested that patients with high *TUBA1C* expression had a poor overall survival rate.

### 3.4 High *TUBA1C* expression is an independent risk factor for OS of breast cancer patients

The Kaplan-Meier curve showed that the OS of breast cancer patients with high *TUBA1C* expression was lower than that of patients with low *TUBA1C* expression ($P < 0.05$) (S1 Fig). To further validate this result, the correlations between different expression levels of *TUBA1C* and the OS, PPS, RFS, and DMFS of breast cancer patients were analyzed using the Kaplan-

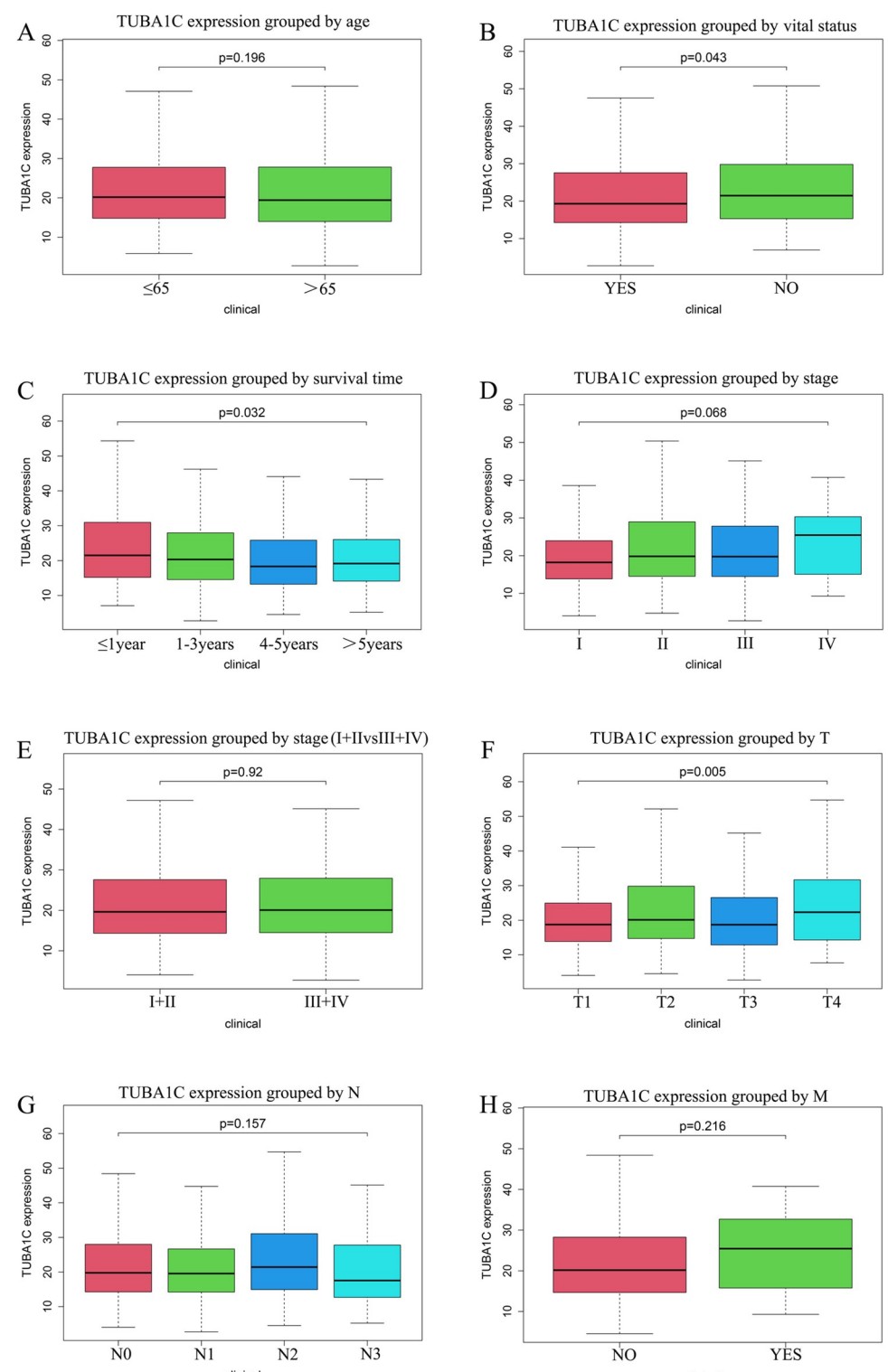

**Fig 3. *TUBA1C* expression in breast cancer patients.** A. Age. B. Vital status. C. Survival time. D-E. Stage. F. T classification. G. N classification. H. M classification.

**Table 2. Correlation between expression of *TUBA1C* and clinicopathological features of patients with breast cancer (logistic regression).**

| Clinical characteristics | Total (N) | Odds ratio in TUBA1C expression | p-Value |
|---|---|---|---|
| Age (continuous) | 1085 | 0.965(0.737–1.264) | 0.798 |
| Stage (I vs. IV) | 202 | 0.972(0.517–1.882) | 0.059 |
| Status (tumor free vs. with tumor) | 1084 | 1.517(1.060–2.182) | 0.023 |
| Distant metastasis (positive vs. negative) | 924 | 1.644(0.686–4.190) | 0.274 |
| Lymph nodes (positive vs. negative) | 1066 | 0.949(0.719–1.252) | 0.712 |

The classification of dependent variables is carried out by using the median expression level to distinguish high and low expression groups.

Meier Plotter (an online database). The results showed that compared with patients with low *TUBA1C* expression, patients with high *TUBA1C* expression had a worse OS (P = 0.0088), RFS (P = 4.3e-07), and DMFS (P = 2.2e-05), and a significantly higher PPS (P = 0.0024) (Fig 4A–4D).

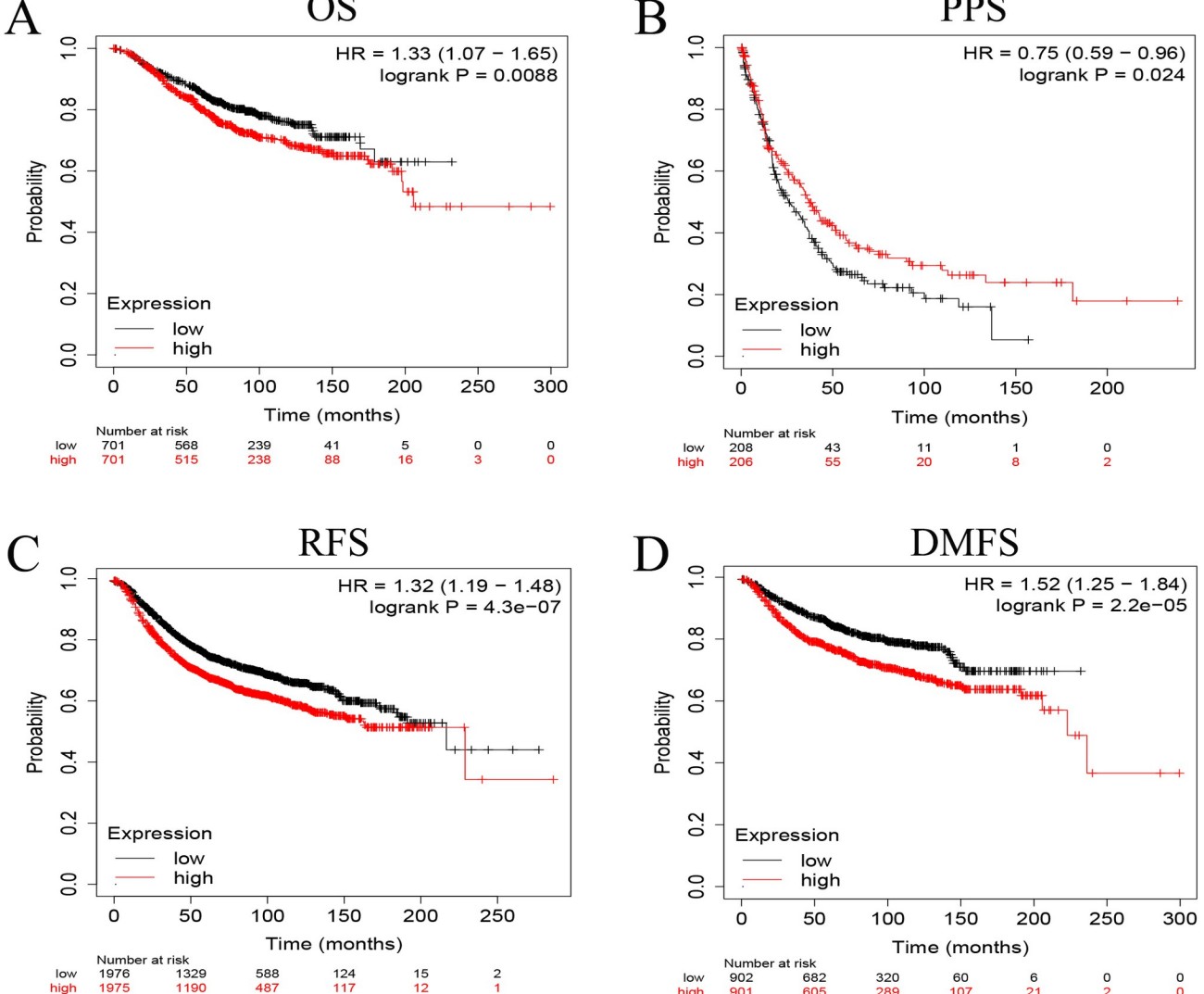

**Fig 4. Kaplan-Meier Plotter in breast cancer patients.** (A-D) Correlation analysis between the different expression levels of *TUBA1C* and OS, PPS, RFS, and DMFS of breast cancer patients.

**Table 3. Univariate analysis of overall survival in *TUBA1C* expression.**

| Parameter | Univariate analysis | | |
|---|---|---|---|
| | HR | 95% CI | P value |
| Age | 1.0337 | 1.0188–1.0489 | 0.0000 |
| Stage | 2.0532 | 1.6146–2.6109 | 0.0000 |
| Tumor | 1.4807 | 1.1840–1.8518 | 0.0006 |
| Node | 1.6680 | 1.3845–2.0097 | 0.0000 |
| Metastasis | 6.1217 | 3.3606–11.1512 | 0.0000 |
| TUBA1C | 1.0193 | 1.0067–1.0321 | 0.0026 |

**Table 4. Multivariate survival model after variable selection.**

| Parameter | Multivariate analysis | | |
|---|---|---|---|
| | HR | 95% CI | P value |
| Age | 1.0364 | 1.0211–1.0520 | 0.0000 |
| Stage | 1.4893 | 0.8687–2.5533 | 0.1475 |
| Tumor | 1.0399 | 0.7580–1.4266 | 0.8084 |
| Node | 1.3002 | 0.9586–1.7634 | 0.0914 |
| Metastasis | 1.4017 | 0.5981–3.2850 | 0.4371 |
| TUBA1C | 1.5430 | 1.1927–1.9962 | 0.0010 |

In addition, univariate and multivariate Cox analyses also showed that high *TUBA1C* expression was an independent risk factor for OS in breast cancer patients (hazard ratio [HR] = 1.5430, 95% confidence interval [CI]: 1.1927–1.9962, P = 0.0010) (Tables 3 and 4).

## 3.5 *TUBA1C*-correlated signaling pathways by GSEA

To determine the signaling pathways that were differentially activated by *TUBA1C* in breast cancer, GSEA was performed between datasets with low and high *TUBA1C* expression. GSEA determined significant differences (FDR < 0.25, NOM p < 0.05) in enrichment in the molecular signatures database (MSigDB) (c2.cp.kegg.v6.2.symbols.gmt). Based on the NES, the most significantly enriched signaling pathways are listed in Table 5 and Fig 5A–5F. The GSEA showed that the high *TUBA1C* expression phenotype was differentially enriched in cell cycle, basal transcription factors, P53 signaling pathway, pathways in cancer, TOLL-like receptor signaling pathway, and NOD-like receptor signaling pathway.

**Table 5. Gene set enriched in high *TUBA1C* expression phenotype.**

| Gene set name | NES | NOM p-val | FDR q-val |
|---|---|---|---|
| KEGG_CELL_CYCLE | 2.52 | 0.000 | 0.000 |
| KEGG_PATHWAYS_IN_CANCER | 1.71 | 0.014 | 0.049 |
| KEGG_P53_SIGNALING_PATHWAY | 2.00 | 0.000 | 0.010 |
| KEGG_BASAL_TRANSCRIPTION_FACTORS | 2.17 | 0.000 | 0.001 |
| KEGG_NOD_LIKE_RECEPTOR_SIGNALING_PATHWAY | 1.79 | 0.023 | 0.034 |
| KEGG_TOLL_LIKE_RECEPTOR_SIGNALING_PATHWAY | 1.77 | 0.018 | 0.040 |

NES: Normalized enrichment score; NOM: Nominal; FDR: False discovery rate. Gene sets with NOM P-value <0.05 and FDR q-value <0.25 were considered as significant.

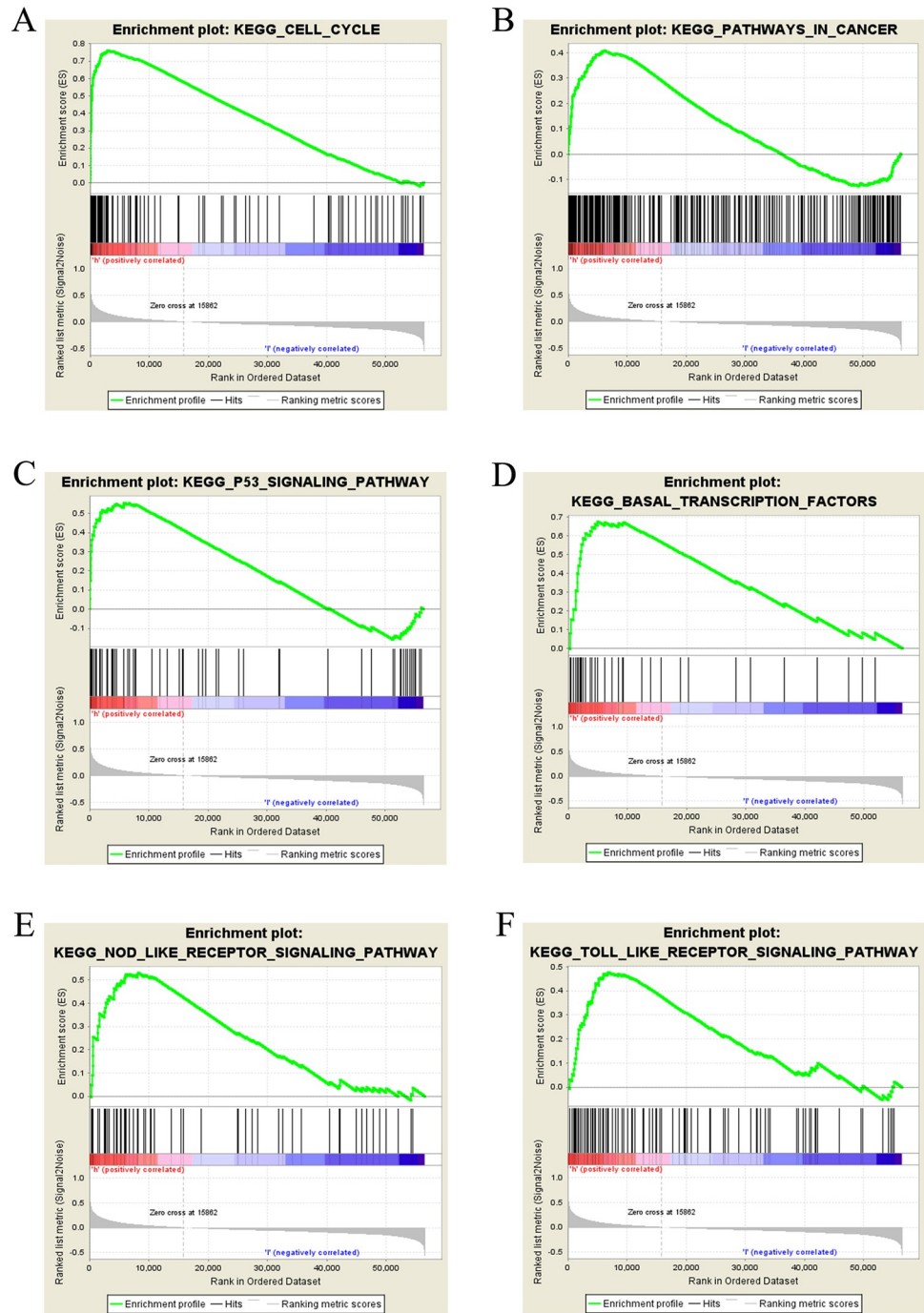

**Fig 5. Enrichment plots from GSEA.** The results of GSEA showed that (A) cell cycle, (B) pathways in cancer, (C) P53 signaling pathway, (D) basal transcription factor, (E) TOLL-like receptor signaling pathway and (F) NOD-like receptor signaling pathway in breast cancer with high expression of *TUBA1C* were differentially enriched.

## Discussion

Many previous studies have confirmed that *TUBA1C* can play a role in the occurrence and development of a variety of tumors. Related research reports confirm that *TUBA1C* can promote liver cancer cell proliferation and invasion and can be used as a prognostic indicator in

liver cancer [12]. Mugahed Abdullah Hasan Albahde et al. reported that high *TUBA1C* expression could affect the cell cycle of pancreatic cancer cells and promote the occurrence and progression of pancreatic cancer [10]. In addition, *TUBA1C* can also negatively regulate miR-143-3p to promote the proliferation of lung cancer cells and reduce cancer cell apoptosis [11]. However, few studies on *TUBA1C* in breast cancer have been published. Although other studies have included *TUBA1C* in the screening of differentially expressed genes in breast cancer, the focus of these studies was not *TUBA1C*, and thus, the correlation between *TUBA1C* and the prognosis of breast cancer patients has not been thoroughly studied [19]. In this study, the expression and potential prognostic value of *TUBA1C* in breast cancer were investigated.

In this study, analysis of high-throughput RNA-seq data from the TCGA database confirmed that *TUBA1C* expression in breast cancer tissues was significantly higher than that in normal tissues, and the pairwise difference analysis between cancer and paired paracancerous tissues also showed that in cancer tissues, *TUBA1C* expression was significantly higher than that in paired paracancerous tissues. According to previous research, high *TUBA1C* expression can promote cell proliferation, which may lead to the poor prognosis of breast cancer patients. In addition, the abnormally high *TUBA1C* expression in breast cancer tissues was closely related to survival status, survival time, and tumor size. Compared with the low expression group, patients with high *TUBA1C* expression had worse OS and were more prone to disease recurrence; moreover, DMFS was also significantly lower. Univariate and multivariate Cox analyses also showed that high *TUBA1C* expression was an independent risk factor for OS in breast cancer patients.

To further investigate the function of *TUBA1C* in breast cancer, the pathways related to *TUBA1C* were analyzed by GSEA. The results showed that the high *TUBA1C* expression phenotype was differentially enriched in cell cycle, basal transcription factor, P53 signaling pathway, pathways in cancer, TOLL-like receptor signaling pathway, and NOD-like receptor signaling pathway. The correlations between *TUBA1C* and the TOLL-like receptor signaling pathway and the NOD-like receptor signaling pathway in breast cancer were first reported in this study, but the specific regulatory mechanisms remain to be further elucidated.

Analysis of the HPA database showed that the upregulation of *TUBA1C* expression in lung cancer, liver cancer, pancreatic cancer, and breast cancer tissues exhibited the same trend. Studies targeting lung cancer, liver cancer, and pancreatic cancer have all confirmed that *TUBA1C* promotes tumorigenesis and progression and is a potential prognostic biomarker for liver cancer and pancreatic cancer.

In summary, a comprehensive bioinformatics analysis using the TCGA database was performed, and the results showed that *TUBA1C* is a potential biomarker that predicts a poor prognosis in breast cancer. However, the study design had some limitations, and additional prospective studies and experiments are still needed to reveal the biological function of *TUBA1C* in breast cancer.

## Supporting information

**S1 Fig. Correlation analysis between the different expression levels of *TUBA1C* and OS.**
The data were processed using R software (version 4.0.3).
(TIF)

## Acknowledgments

The authors express our sincerely acknowledgment to Mr. Zhang Chengwu who provided assistance in study design.

## Author Contributions

**Conceptualization:** Miaozhou Wang, Xingfa Huo.

**Investigation:** Jiarui Du, Zhen Liu.

**Methodology:** Fuxing Zhao, Dengfeng Ren.

**Supervision:** Wenwen Wang, Qiqi Xie.

**Validation:** Jinming Li.

**Visualization:** Jiuda Zhao, GuoShuang Shen.

**Writing – original draft:** Yi Zhao.

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
