## [Decision Letter · Decision Letter 0]

30 Dec 2021

PONE-D-21-07123

Elevated expression of TUBA1C in breast cancer predicts poor prognosis

PLOS ONE

Dear Dr. zhao,

Thank you for submitting your manuscript to PLOS ONE. After careful consideration, we feel that it has merit but does not fully meet PLOS ONE’s publication criteria as it currently stands. Therefore, we invite you to submit a revised version of the manuscript that addresses the points raised during the review process.

We look forward to receiving your revised manuscript.

Kind regards,

Suhwan Chang

Academic Editor

PLOS ONE

Journal Requirements:

"All named authors agree with the submission and declare that there were no conflicts of interest regarding the publication of this manuscript."

Additional Editor Comments (if provided):

Please follow the comments from the reviewer carefully.

Reviewers' comments:

Reviewer's Responses to Questions

**Comments to the Author**

1. Is the manuscript technically sound, and do the data support the conclusions?

Reviewer #1: Yes

2. Has the statistical analysis been performed appropriately and rigorously? 

Reviewer #1: I Don't Know

3. Have the authors made all data underlying the findings in their manuscript fully available?

Reviewer #1: Yes

4. Is the manuscript presented in an intelligible fashion and written in standard English?

Reviewer #1: Yes

5. Review Comments to the Author

Reviewer #1: This appears to be an interesting study but the foolowig changes in the manuscript are suggested.

1. There are fomatting issues, formatting should be according to PLOS guidelines. (See lastline of first paragraph in the Background section).

2. Cite references considering the PLOS ONE guidelines. (See second line of Discussion section).

6. PLOS authors have the option to publish the peer review history of their article (what does this mean?). If published, this will include your full peer review and any attached files.

Reviewer #1: No

---

## [Author Response · Author response to Decision Letter 0]

17 Jan 2022

Reviewer #1: This appears to be an interesting study but the foolowig changes in the manuscript are suggested.

1. There are fomatting issues, formatting should be according to PLOS guidelines. (See last line of first paragraph in the Background section).

Response: Thank you for the suggestion. We have modified the format of the manuscript according to the requirements stated in the PLOS ONE guidelines.

2.Cite references considering the PLOS ONE guidelines. (See second line of Discussion section).

Response: Thank you for the suggestion. We have modified the format of the references according to the requirements stated in the PLOS ONE guidelines.

---

## [Decision Letter · Decision Letter 1]

25 May 2022

Elevated expression of TUBA1C in breast cancer predicts poor prognosis

PONE-D-21-07123R1

Dear Dr. Yi Zhao,

We’re pleased to inform you that your manuscript has been judged scientifically suitable for publication and will be formally accepted for publication once it meets all outstanding technical requirements.

Kind regards,

Suhwan Chang

Academic Editor

PLOS ONE

Additional Editor Comments (optional):

No further comments.

Reviewers' comments:

Reviewer's Responses to Questions

**Comments to the Author**

1. If the authors have adequately addressed your comments raised in a previous round of review and you feel that this manuscript is now acceptable for publication, you may indicate that here to bypass the “Comments to the Author” section, enter your conflict of interest statement in the “Confidential to Editor” section, and submit your "Accept" recommendation.

Reviewer #2: All comments have been addressed

2. Is the manuscript technically sound, and do the data support the conclusions?

Reviewer #2: Yes

3. Has the statistical analysis been performed appropriately and rigorously? 

Reviewer #2: Yes

4. Have the authors made all data underlying the findings in their manuscript fully available?

Reviewer #2: Yes

5. Is the manuscript presented in an intelligible fashion and written in standard English?

Reviewer #2: Yes

6. Review Comments to the Author

Reviewer #2: Authors have done a good job in finding TUBA1C as potential biomarkers in breast cancer prediction.

For purpose of repeating such analysis by wider community, it would be essential if authors can share the code used in doing the analysis. Submitting it on GitHub or similar resource and sharing the link in the paper of the same should be sufficient.

Also a flow chart of analysis pipeline as initial figure would be helpful.

Authors need to provide more info on choice of method used for differential expression analysis and if there's any covariates and cofactors of concern based on clinical info available.

7. PLOS authors have the option to publish the peer review history of their article (what does this mean?). If published, this will include your full peer review and any attached files.

Reviewer #2: No

---

## [Editor Report · Acceptance letter]

30 May 2022

PONE-D-21-07123R1 

Elevated expression of TUBA1C in breast cancer predicts poor prognosis 

Dear Dr. Zhao:

I'm pleased to inform you that your manuscript has been deemed suitable for publication in PLOS ONE. Congratulations! Your manuscript is now with our production department. 

Kind regards, 

on behalf of

Dr. Suhwan Chang 

Academic Editor

PLOS ONE